# Enhancing the Mechanical Properties of Waterborne Polyurethane Paint by Graphene Oxide for Wood Products

**DOI:** 10.3390/polym14245456

**Published:** 2022-12-13

**Authors:** Dandan Xu, Guotao Liang, Yanran Qi, Ruizhi Gong, Xingquan Zhang, Yumin Zhang, Baoxuan Liu, Linglong Kong, Xiaoying Dong, Yongfeng Li

**Affiliations:** 1College of Forestry, State Forestry and Grassland Administration Key Laboratory of Silviculture in Down-Stream Areas of the Yellow River, Shandong Agricultural University, Taian 271018, China; 2Shandong Laucork Development Co., Ltd., Jining 272100, China

**Keywords:** waterborne polyurethane, graphene oxide, mechanical properties, coating, wood products

## Abstract

Water-based polyurethane paint is widely used for wood furniture by virtue of the eco-friendliness, rich gloss, and flexible tailorability of its mechanical properties. However, its low solution (water or alcohol) resistance and poor hardness and wear resistance limit its application. The emerging graphene oxide has a high specific surface area and abundant functional groups with excellent mechanical properties, endowing it with great potential to modify waterborne polyurethane as a nanofiller. In this study, graphene oxide prepared by Hummers’ method is introduced in the chemosynthetic waterborne polyurethane through physical blending. The testing results demonstrate that the appropriate usage of graphene oxide at 0.1 wt% could obviously improve water absorption resistance and alcohol resistance, significantly enhancing the mechanical properties of waterborne polyurethane paint. The corresponding tensile strength, abrasion resistance, and pendulum hardness of the graphene oxide-modified paint film increase by 62.23%, 14.76%, and 12.7%, respectively, compared with the pristine paint film. Meanwhile, the composite paint film containing graphene oxide possesses superiority, including gloss, abrasion resistance, pendulum hardness, and tensile strength in contrast with the commercial paint. The use of graphene oxide to enhance the waterborne polyurethane possesses strong operability and practical value, and could provide useful reference for the modification of waterborne wood paint.

## 1. Introduction

The released volatile organic compounds (VOCs) from the traditional solvent-based wood coatings could cause environmental pollution and impair people’s health [1,2,3,4]. Therefore, environmentally friendly water-based wood coatings with low VOCs-release are becoming a research hotspot in this field [5]. Water-based polyurethane (WPU) paint has shown great development potential in wood furniture coating due to its environment-friendliness, rich gloss, strong adhesion, and excellent water and abrasion resistance [6,7,8]. However, its mechanical properties and durability are poor compared with traditional solvent-based wood coatings, which affects its wide application in wood products [9,10,11].

Researchers have tried to improve the mechanical properties, water resistance, and solvent resistance of WPU by cross-linking copolymerization to convert linear or lightly branched macromolecules into a three-dimensional structure [12,13]. Li et al. [14] prepared a hybrid coating of polysiloxane polyurethane molecules by introducing trimethylolpropane and aqueous polycarbodiimide as cross-linkers in the prepolymer to construct a double cross-linked structure, which could improve the ductility and toughness of the composite paint film. Sun et al. [15] used flame retardant monomer (bis (2-hydroxyethyl) aminomethyl phosphate diethyl ester), cross-linking monomer (dihydroxyacetone), and small molecules (adipic acid dihydrazide) to produce the self-cross-linked fluorinated flame retardant WPU, which effectively improved the tensile and flexural properties of the paint film. However, cross-linking copolymerization always requires precise synthesis technology and appropriate cross-linking agents. Adding nanofillers is also an effective and simple method to enhance the mechanical properties of WPU. Nanofillers could strongly interact with the polymer matrix due to their small size effect, surface interface effect, and other properties, and provide a good modification effect on polyurethane materials. Zero-dimensional, one-dimensional, or two-dimensional inorganic nanomaterials, such as SiO_2_ [16], TiO_2_ [17], ZnO [18], flake clay [19], and hydroxyapatite nanoparticles [20], could be introduced into the organic resin matrix to enhance the hardness, wear resistance, Young’s modulus, water resistance, and weather resistance of the coatings. Chen et al. [21] improved the antibacterial properties and adhesion of WPU coatings by employing nanocellulose crystals (NCC) and silver nanoparticles (AgNPs). However, introducing some inorganic fillers may lead to the increasing brittleness of the coating film, even making it difficult to meet the practical requirements [22,23]. Therefore, selecting suitable nanofillers could be key to preparing a high-performance WPU for wood coating [24].

Graphene oxide (GO) possesses a single-layer honeycomb-like lattice structure with sp^2^ carbon atoms, corresponding to a huge specific surface area and excellent mechanical properties [25,26], which could guarantee the mechanical enhancement and avoid the increased brittleness of WPU compared with the common inorganic fillers at a low dosage. In addition, abundant hydrophilic groups endow GO with benign hydrophilicity and high reactivity, which could also act as active sites for modification to broaden the application of GO [27,28,29]. GO could integrate with the polyurethane (PU) through the interaction between functional groups and further enhance the modification effect when improving the properties of WPU. Sinh et al. [30] introduced isocyanate-functionalized graphene (iGO) into thermoplastic PU and improved the mechanical and thermal properties of PU/iGO composites. Wan et al. [31] developed a series of self-healing aqueous PU/graphene oxide nanocomposites with good dispersion stability and thermal stability. Therefore, GO holds great potential for improving the aqueous PU as the nanofiller. In this study, GO is added into aqueous PU coatings by physical blending to prepare the composite coatings. A small dosage of GO could obviously improve the properties of the composite paint film and render the composite paint superior to the commercial paint. The corresponding tensile strength could increase by 62.23%, while the abrasion resistance and pendulum hardness of the GO-modified paint film increase by 14.76% and 12.7%, respectively, compared with the pristine paint film.

## 2. Materials and Experiments

### 2.1. Materials

Isophorone diisocyanate (IPDI; 99%) was obtained from Jining Hongming Chemical Reagent Co., Ltd.( Jining, China). Polypropylene glycol 2000 (PPG2000; 99%) was provided by Qingdao Usolf Chemical Technology Co., Ltd.(Qingdao, China). Dihydroxymethylpropionic acid (DMPA; 99%) was supplied by Shanghai Macklin Biochemical Technology Co., Ltd. (Shanghai, China). We purchased 1,4-butanediol (BDO; 99%), N-methylpyrrolidone (NMP; 99%), and acetone (99%) from Taian Keshan Biotechnology Co., Ltd. (Taian, China. Dibutyltin dilaurate (DBTDL; 98%) was produced by Tianjin Damao Chemical Reagent Factory., China. Triethylamine (TEA; 99%) was acquired from Shandong Xiya Chemical Industry Co., Ltd. (Linyi, China). Graphite powder (300 mesh) was provided by Qingdao Henglide Graphite Co., Ltd. (Qingdao, China). All the reagents were directly used without further purification.

### 2.2. Preparation of GO, Composite Emulsions, and Paint Film

Graphene oxide was prepared from graphite powder by using the Hummers’ method in Figure 1. Graphite powders (1 g) were put into a beaker with concentrated sulfuric acid (40 mL) and sodium nitrate (0.3 g). Potassium permanganate (7 g) was slowly added and stirred in an ice-water bath for 30 min. The temperature was slowly increased to 35 °C and maintained for 2 h, then distilled water (115 mL) was added with continuous stirring. Subsequently, the temperature was raised to 95 ± 3 °C and maintained for 30 min, then hydrogen peroxide solution (10 mL, 30 wt%) was added to terminate the reaction with no bubbling, producing the golden yellow solution. After that, the supernatant was removed by standing and stratifying, and the residual solution was washed with diluted hydrochloric acid (~1 mol L^−1^) and deionized water for several times to pH 7 by centrifuging at 8000 r min^−1^, followed by the dialysis in deionized water for 7 days. Finally, the uniformly dispersed GO gel suspension was obtained by adding moderately deionized water and ultrasonic treatment.

The synthesis of WPU was mainly divided into three stages, as shown in Figure 2. The first stage was prepolymerization reaction. IPDI was added in a four-neck flask with PPG-2000 and reacted at 65 °C for 2 h to form prepolymer I. The second stage was chain expansion reaction. The mixed BDO with a certain amount of acetone was continually reacted with prepolymer I for 2 h at 65 °C, then DMPA in 20 mL NMP was dropwise added and stirred at 70~80 °C for 2 h to synthesize prepolymer III. The third stage was the neutralization reaction. TEA was added to neutralize the reaction for 0.5 h at 40 °C to obtain aqueous PU prepolymer. The GO suspension was slowly added dropwise to the prepolymer using a peristaltic pump at a dropwise acceleration of 10 mL min^−1^ and stirred at 800 rpm for 1 h. The composite emulsion was then sonicated for 0.5 h. Finally, the homogeneously dispersed composite emulsion was vacuum-treated for 1 h. The purpose was to remove excess air bubbles from the emulsion to prepare the GO-modified WPU composite emulsion (labeled as GO-WPU). The mass of GO in GO-WPU was controlled at 0 wt%, 0.05 wt%, 0.1 wt%, 0.4 wt%, 0.7 wt%, and 1 wt% based on the solid content of aqueous PU.

The composite paint film was manufactured by pouring a certain amount of the prepared emulsion into a Petri dish, air-drying, and drying in an oven at a certain temperature. The GO film was made in the same method. The tested wood samples were prepared by spraying sealer primer and primer twice on virgin maple veneer, cleaning with sandpaper, and finally spraying the prepared GO-WPU paint.

### 2.3. Structural Characterization

The microscopic morphology of GO was observed using an atomic force microscope (AFM; FM-Nanoview1000, Suzhou FMSM Precision Instruments Co., Ltd., Suzhou, China) by diluting GO suspension, dropping it onto the mica flakes, and natural drying. The tested films were embrittled by using liquid nitrogen and observed at different magnifications by employing a field emission scanning electron microscope (FE-SEM; S4800, Hitachi, Ibaraki, Japan) with an accelerating voltage of 10.0 kV. The chemical composition of GO and the composite paint film was investigated by employing Fourier transform infrared spectroscopy (FTIR; IRTracer-100, Shimadzu, Japan) and UV-Vis-NIR Raman spectrometer (Raman; LabRAM HR800, HORIBA Jobin Yvon, France). FTIR was performed using ATR mode with a resolution of 2 cm^−1^ and 32 scans. The crystalline structures of the paint films were confirmed by adopting X-ray diffraction (XRD; D8 Advance, Bruker, Germany). Thermogravimetric analysis (TGA; SDT Q500, TA, USA) was used to test the thermal stability of the composite paint film in the range of 30~800 °C with a heating rate of 10 °C min^−1^ and N_2_ atmosphere.

### 2.4. Performance Testing

The wood boards sprayed with WPU were tested for glossiness with a three-angle gloss tester (GZ-II) with a selected incidence angle of 60° as per GB/T 4896.6-2013. The test was conducted after calibration and the instrument and board were pressed during the test to avoid inaccurate test results due to light leakage. The adhesion of the paint film was measured by using a coating scribe (BGD 504/1) in accordance with GB/T 4893.4-2013. The abrasion resistance of the paint film was examined by adopting a paint film abrasion meter (BGD523) based on GB/T1768-2006. Pendulum hardness of the paint film was evaluated by managing a double pendulum hardness tester (BGD 508), which complied with GB/T1730-1993. The test was performed by first adjusting the double pendulum hardness tester so that the number of pendulum swings reached the standard, and then the test was conducted. The pencil hardness of the paint film was surveyed by utilizing a hand-cranked pencil hardness meter as per GB/T6739-2006. The water resistance, alcohol resistance, and alkali resistance of the paint film were confirmed according to GB/T 4893.1-2005. The dry time of the paint film was tested by operating a dry time tester (BGD 263) as per GB/T 1728-1979. The coating was sanded with 400# sandpaper based on GB23999-2009. The tensile strength and elongation at break were tested using a universal mechanical testing machine in terms of GB/T1040-1992. All the test results were averaged over 3 times.

## 3. Results and Discussion

GO has a huge specific surface area and excellent mechanical properties, reactivity, optical properties, and electrical conductivity. The prepared GO by the Hummers’ method comprises micron-sized sheets with some folds and stacking (Figure 3a,b), while the graphene has a thickness of 0.335 nm and is layered in sheets [32]. Graphene starts to decompose rapidly at 500 °C and decomposes completely at about 600 °C, owing to the burning of carbon [33]. Three significant mass losses occurred for GO in the tested temperature range (Figure 3c). The mass loss below 165 °C is mainly caused by the volatilization of the adsorbed water, and the mass loss at 165~200 °C is due to the thermal decomposition of oxygen-containing groups in GO, indicating the abundant surface functional groups of the obtained GO. Finally, the mass loss at 200~660 °C is mainly due to the combustion of the carbon skeleton. The absorption peak of graphene at 1630 cm^−1^ is attributed to the stretching vibration of the C=C peak in the sp^2^ structure, and the small and narrow -OH stretching vibration peak (~3437 cm^−1^) may be caused by the slight oxidation of its surface layer in the FTIR spectrum (Figure 3d). The main absorption peak of GO at 3340 cm^−1^ refers to the stretching vibration of the O-H group, and the characteristic peaks at 1730 cm^−1^ and 1630 cm^−1^ belong to the C=O stretching of the carboxyl and/or carbonyl functional groups displayed in Figure 3d. The peaks located at 1226 cm^−1^ and 1044 cm^−1^ are assigned to C-O stretching vibrations [34], indicating that GO is rich in oxygen-containing functional groups. The Raman spectrum of graphene shows the typical D band, G band, and 2D band [35]. The D band reflects the disorder of graphene due to motifs, edges, and defects. The G band is related to the crystal sp^2^ carbon atoms in the carbon ring from the highly ordered carbon network plane of the graphite layer. The 2D band is a double-resonance process involving two phonons of opposite momentum, and could be used to characterize the number of graphene layers (Figure 3e) [35]. The typical peaks of GO in the Raman spectrum (Figure 3e) at ~1588 cm^−1^, ~1345 cm^−1^, and ~2800 cm^−1^ belong to the G band, D band, and G band, respectively [35,36]. The G band of GO shifts to higher wavenumbers, which are related to the formation of sp^3^ carbon atoms during the oxidation process of graphite, while the in-plane sp^2^ region decreases, corresponding to the broadening D band of GO. The high value of I_D_/I_G_ (1.12) and the obvious G band reflect the certain disordered degree and defects in GO. The abundant oxygen-containing functional groups and the existing defects could endow GO to be uniformly dispersed in aqueous PU and have strong interactions with PU, further exerting the positive influence on the performance of PU composites.

The added GO could be uniformly distributed in the WPU film without obvious agglomerates as shown in Figure 3f, indicating that GO has good compatibility with WPU [37]. The WPU lacquer film shows certain crystallization, with typical characteristic diffraction peaks among 5~20 °. By introducing 0.1 wt% GO, the peak intensity of the WPU film occurs with apparent variation (Figure 3g). The peak at ~18.7 ° becomes the strongest peak, indicating changes to the preferential crystal plane from (001) to (110), which may be caused by the interaction between GO and WPU chain segments along with the influence on the movement and regularity of the chain segments. The effect on the crystal structure of WPU film may also endow it with good mechanical properties. The broad characteristic peak at ~3350 cm^−1^ is assigned to N-H vibrations (Figure 3h), accompanied by a slight shift that is probably due to the formation of a hydrogen bond with the addition of 0.1 wt% GO. The enhancement in the characteristic peaks of -CHO (~2800 cm^−1^) and C-H (~2850 cm^−1^) indicates the hydrogen bond between GO and the aqueous PU, which may positively impact the mechanical properties of aqueous PU. The TG-DTG curves of both tested samples showed similar phenomena (Figure 3i). A slight weight loss (~5 wt%) first occurred in the range from 50 to 220.6 °C, owing to water evaporation. Mass loss is then accelerated among 279.5~419.4 °C, accompanied by a weight loss of ~80% during the decomposition of WPU [38]. Adding GO endows the composite paint film a higher temperature at the fastest decomposition rate. These results demonstrate that GO has good compatibility with WPU and may contribute to improving the performance of the composite paint film.

The mechanical properties of PU materials play an important role in practical applications. The effect of GO on the performance of the hybrid paint is carefully studied through a series of performance tests(Table 1). The pencil hardness of the composite film increases from 3B to 2B as the GO addition is above 0.1 wt%, which means the addition of GO has a positive effect on the abrasion resistance and compressive strength of the paint film. This elevation in pencil hardness may be ascribed to the interaction between GO and resin molecules and the affected cross-linking degree. The sand ability and alkali resistance of the paint films show no significant difference, displaying the same performance levels. [39,40]. However, the water resistance and alcohol resistance of the composite coating exhibit promotion as the added GO is not lower than 0.1wt%. The water resistance of the paint film is improved from Grade 3 to 2. Thus, water molecules are difficult to penetrate in the paint film, which could inhibit the fade, crack, blister, and float of the coating in a humid environment. Similarly, the alcohol resistance of the composite film could be improved from Grade 5 to 4. The enhanced solvent resistance of the composite paint film is mainly due to the interaction between GO and PU molecules, as well as the extended pathway for the movement of ions or molecules within the PU by GO additives [41,42,43].

Figure 4a shows the glossiness of GO-modified PU with different concentrations, and they significantly decrease with the increased amount of GO, owing to the existing GO particles on the surface of the solid coating matrix as the light-scattering points [44]. Thus controlling the additive amount of GO in a rational level is essential. Surface drying time and hard drying time could affect the practical application of the paint on wood products. The difference between the surface drying time and hard drying time of the tested paints is small (Figure 4b,c), and are mainly centered on 30 ± 5 min and 58 ± 5 min, respectively. These values reach the minimums with the GO dosage of 0.1 wt%, and gradually increase as GO addition is above 0.1 wt%. Thus, adding 0.1 wt% GO in WPU could facilitate the evaporation of water during the drying process, which further enhances the operability in the application process.

Obvious differences in the adhesion of the tested paint films are not observed because the dosage of GO is below 0.4 wt% (Figure 4d). However, the corresponding adhesion reduction occurs with the increasing amount of GO in the composite paint. Excessive GO in the composite paints may lead to inhomogeneous dispersion, affecting the two-phase interfacial interaction between the resin matrix and the substrate surface, accompanied by the relatively weak adhesion [45]. The wear resistance of the coating could be obviously improved with a small mass-loss rate by introducing GO (Figure 4e), owing to the enhanced interaction between PU molecules and GO. During the process of mechanical wearing, the added GO with large specific surface area could enhance the intermolecular forces of the resin matrix, and further relieve the damage by the shear and tensile forces. Meanwhile, GO could also promote the pendulum hardness of the composite paint films with the increase in GO concentration (Figure 4f). The peak increment reaches 12.7% compared with the GO-free WPU film by enhancing the interaction among the resin molecules to bear more external pressure [46].

The tensile strength of paint film could significantly influence its application in wood products. Adding a small amount of GO in WPU could distinctly increase the tensile strength of the composite paint films (Figure 4g), which may be induced by the hydrogen bonding between the oxygen-containing groups on GO and the hydroxyl groups on PU molecular chains. The peak value could be 6.1 MPa at 0.7 wt%, with an increase in proportion of 62.23%. The elastic properties of monolayer GO could be tested by AFM, and the average Young’s modulus of the monolayer GO is 207.6 ± 23.4 GPa [47]. The relevant value would grow along with the increase in the GO layer. According to the “Tsai-Wu” and “Checkerboard” micromechanics models, the elastic modulus of GO changes positively with the increase in tensile strength in a certain range, implying an enhancement in the elastic properties of the composite [48,49,50,51]. In the meantime, the elongation at the break of the tested paint films appears to display the opposite rule with the increasing dosage of GO. Excessive GO would impair the toughness of the composite paint films (Figure 4h), probably due to the promoted crystallinity of PU through enhancing the intermolecular forces and reducing the slip among the resin molecular chains [52]. The composite film of GO-WPU at 0.1 wt% shows benign elongation at break, with a slight decrease (1.7%) compared with the GO-free paint film. Therefore, the usage of GO at 0.1 wt% could prepare the preferable composite paint with good comprehensive performance (Figure 4i).

Furthermore, the comparison between the prepared paint and commercial water-based paint (marked as CF) is conducted to evaluate the application potential. The related results in Table 2 and Figure 5a demonstrate that the synthesized WPU paints are superior to the CF, especially the modified paint with 0.1 wt% GO. The GO-modified composite paint film possesses the optimal comprehensive performance, which endows its commercial availability for wood products. The specific mechanism of GO in the WPU could be presented in Figure 5b. The surface of GO has abundant oxygen-containing functional groups, which could form hydrogen bonds with the PU molecular chains and facilitate the uniform dispersion of GO in WPU. Then, the originally chain-arranged PU molecular chains may be connected to form a three-dimensional network structure by GO, corresponding to the enhancement in the mechanical properties of WPU [53,54]. Thus, the appropriate dosage of GO could solve the issues of the traditional WPU paint and promote its extensive application on wood products.

## 4. Conclusions

In summary, the composite aqueous PU paint has been obtained by a simple mixing of dispersed GO and PU emulsions. The added GO is demonstrated to significantly improve the mechanical properties and durability of the WPU coating with the optimal usage (0.1 wt%). The corresponding tensile strength, wear resistance, and pendulum hardness could be increased by 62.23%, 14.76%, and 12.7%, respectively, compared with the GO-free sample. Moreover, the prepared composite PU paint exhibits some performance advantages in comparison with the commercial PU paint and great application potentials for wood products. This study focuses on water-based coatings to improve the solid content of water-based coatings and effectively improve the deficiencies of water-based coatings by introducing other organic/inorganic nanomaterials. In the future, we will add GO to water-based coatings by chemical grafting, explore the reaction mechanism, and try to develop UV-curable water-based coatings to broaden the application fields of waterborne coatings. Meanwhile, the effect of GO-modified WPU on constraining termite/microbial/fungal attack could be explored and discussed in future studies.

## Figures and Tables

**Figure 1 polymers-14-05456-f001:**
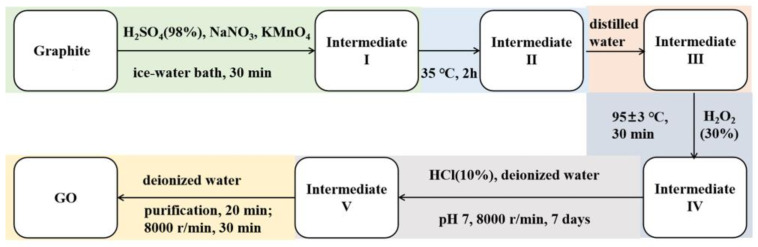
Schematic diagram of GO preparation by Hummers’ method.

**Figure 2 polymers-14-05456-f002:**
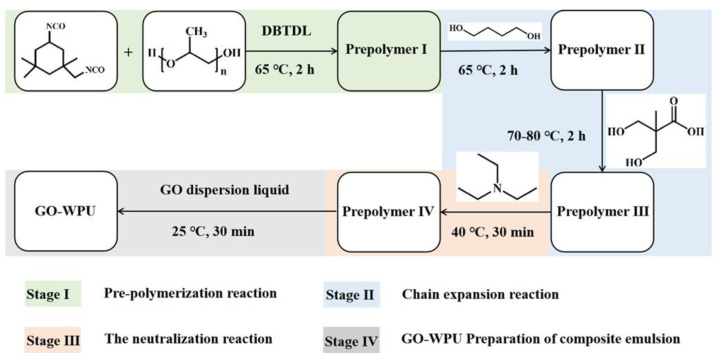
Schematic diagram of oxidized composite waterborne polyurethane.

**Figure 3 polymers-14-05456-f003:**
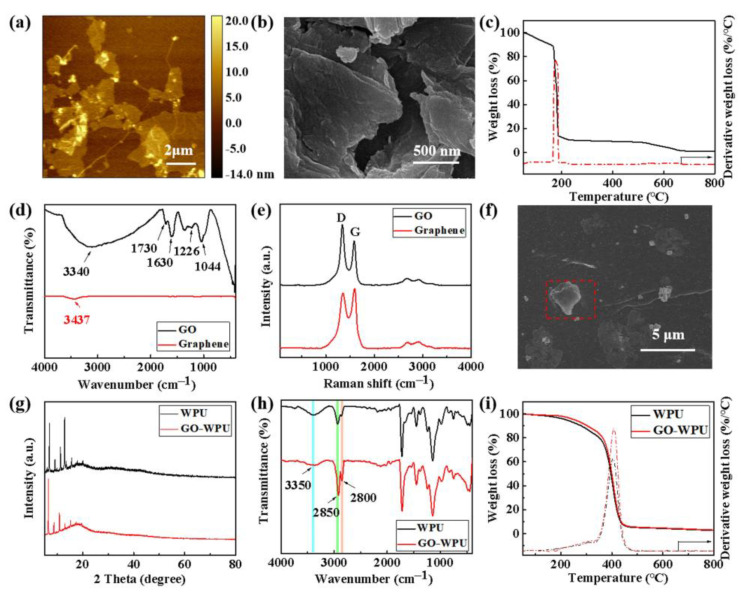
Chemical composition and morphological characterization of graphene, GO, and GO composite paint films. (**a**) AFM image, (**b**) surface morphology, (**c**) TG curves, (**d**) FTIR spectrum and Raman spectrum (**e**) of graphene and GO. (**f**) Cross-sectional image of 0.1 wt% GO-modified WPU paint film. (**g**) XRD patterns, FTIR spectra, (**h**) and (**i**) TG curves of pure and 0.1 wt% GO-modified WPU films.

**Figure 4 polymers-14-05456-f004:**
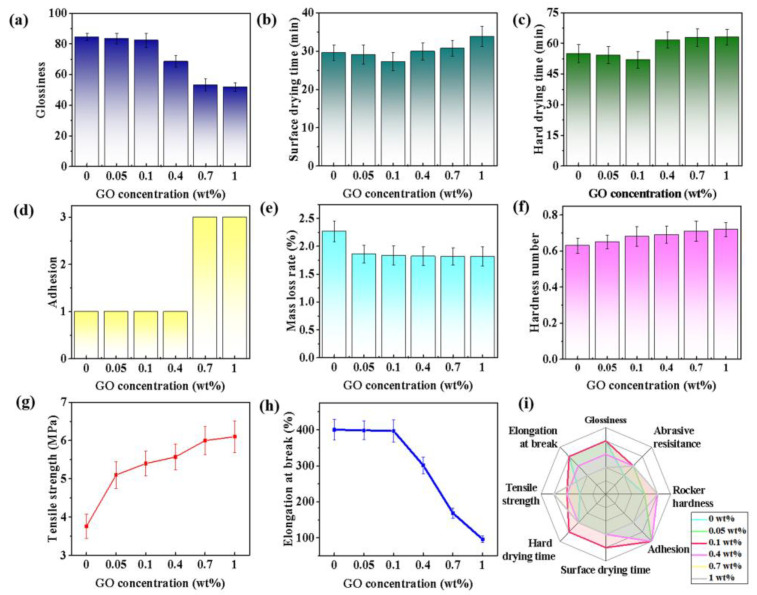
Performance tests of composite paint films, including (**a**) glossiness, (**b**) surface drying time, (**c**) hard drying time, (**d**) adhesion, (**e**) abrasion resistance, (**f**) pendulum hardness, (**g**) tensile strength, and (**h**) elongation at break with different GO concentrations. (**i**) Comprehensive radar map of the GO-WPU with different GO concentrations.

**Figure 5 polymers-14-05456-f005:**
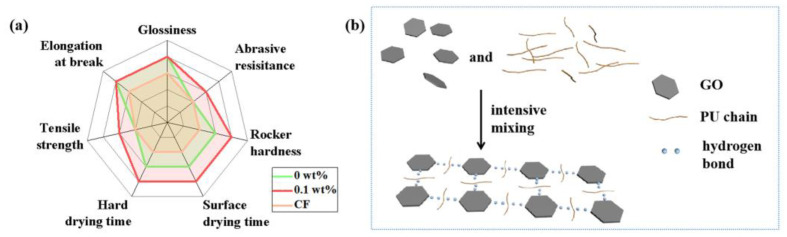
(**a**) Radar diagram of comprehensive performance among the three kinds of WPU films. (**b**) Schematic of GO in WPU to enhance the mechanical properties.

**Table 1 polymers-14-05456-t001:** Properties of GO-WPU paint film with different GO contents.

	Sample	0 wt%	0.05 wt%	0.1 wt%	0.4 wt%	0.7 wt%	1 wt%
Property							
Pencil hardness		3B	3B	2B	2B	2B	2B
Burnish		Easy	Easy	Easy	Easy	Easy	Easy
Hydrolytic resistance		3	3	2	2	2	2
Alcohol resistance		5	5	4	4	4	4
Alkali resistance		1	1	1	1	1	1

Notes: 1 refers to no visible change or no damage. 2 refers to slight visible discoloration, change, or discontinuous imprint only when the light shines on the surface of the sample or is very close to the imprint. 3 refers to slight impression that is visible in several directions, such as nearly complete circle or round mark. 4 refers to serious impression without greatly changed surface structure. 5 refers to serious impression, while the surface structure is changed, the surface material is torn in whole or in part, or the paper is adhered to the test surface.

**Table 2 polymers-14-05456-t002:** The comparison of the performance among the three kinds of WPU films.

	Property	Glossiness	Abrasive Resistance (%)	Rocker Hardness	Surface/Hard Drying Time (min)	Tensile Strength (MPa)
Sample						
0		84.53	2.27	0.63	29.6/55	3.76
0.1		82.1	2	0.65	27.2/52	5.12
CF		73.5	2.793	0.598	33.9/72.57	3.49

## Data Availability

Not applicable.

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
