# Peer review of "Enhancing the Mechanical Properties of Waterborne Polyurethane Paint by Graphene Oxide for Wood Products"

_polymers, 2022, doi:10.3390/polym14245456_

Round 1
Reviewer 1 Report
1a. English language revision required. Authors have used present indefinite tense in result and discussion parts, rather than present perfect tense.
> Page 2, line 96: "Graphite powders (1g) were" should be "Graphite powder (1g) was"
> Page 3, line 98: "under an ice-water bath" should be "in an ice cold water bath"
> Page 5, line 197-198: "The TG-DTG curves of the tested two samples show the similar phenomenon in Figure 3i." should be "TG-DTG curves of both tested samples showed similar phenomenon (Figure 3i)"
>page 5, line 198: "A slight weight loss (~5 wt%) first occurs" should be "A slight weight loss (~5 wt%) first occurred"
> Page8, line 292: "polyurethane paint could be obtained by" should be as "polyurethane paint has been obtained by"
> Page 8, line 292: "simple mixing dispersed" should be as "simple mixing of dispersed"
1b. Water based paint particularly for wood should have been tested for constraining termite/microbial/fungal attack?
2. In section 2.1 Materials: Eflornithine diisocyanate has been abbreviated as IPDI. Correction required.
3. Page 3, line 98: Immediately after ice-water bath treatment, temperature has been stated as 35 degrees.
4. Page 3, line 103-4: Authors statement "washed with ---- to pH 4-5 or neutral" is confusing about the end point. Was it pH 4-5 or neutral (7)?
5. Page 4, line 159: Statement about TGA that two significant mass losses occur for GO in the tested temperature range (Figure 3c), is not right. There are three regions of weight loss.
6. In Figure 3a-e, only GO has been presented, there must also be graphene for a proper comparison, or as an evidence to effective surface modification.
7. Repetition of words may be avoided by using abbreviations, e.g Polyurethane should be replaced by PU
Reviewer 2 Report
Please read and “fully” address the comments listed below:
1. The ABSTRACT is not written in a logical order. Start with an overview of the topic and a rationale for your paper. Describe the methodology you used and the general outline of the manuscript. Also, in the end, state the result in more detail (i.e., provide some numbers).
2. The novelty of your work is still unclear to the reader, which should be further detailed both in the Abstract and Introduction.
3. Please fully explain the process used for mixing the GO with waterborne polyurethane coatings.
4. Increase the size of the texts in Fig. 3.
5. Please fully introduce the elastic properties of GO.
6. Explain in a few sentences how AFM, surface topography, and FTIR tests were done.
7. Please provide more explanation for this sentence: Page 8, Line 279: "Hydrophilic GO could be uniformly dispersed in WPU at the nanoscale.”
8. Explain in detail how glossiness, adhesion, and pendulum hardness were measured, and introduce the setup used for measuring these properties.
9. It is mentioned that by optimal usage of GO (0.1%wt.) the corresponding tensile strength could be increased by 62.23%. This is a valuable statement, which was experimentally determined. However, apart from the experimental analysis, novel micromechanics models have been developed that can be used to estimate the elastic properties of composite-reinforced graphene. In particular, “Tsai-Wu” and “Checkerboard” micromechanics models are commonly used. Therefore, please write a paragraph in your paper introducing these micromechanics models that can be used to estimate the tensile strength of waterborne polyurethane coatings reinforced with GO and reference the papers below (selected from Chen and Aghdam’s research labs).
Tsai-Wu:
· Chen, X., Sun, X., Chen, P., Wang, B., Gu, J., Wang, W., ... & Zhao, Y. (2021). Rationalized improvement of Tsai–Wu failure criterion considering different failure modes of composite materials. Composite Structures, 256, 113120.
· Wang, B., Chen, X., Wang, W., Yang, J., & Zhang, R. (2022). Post-buckling failure analysis of composite stiffened panels considering the mode III fracture. Journal of Composite Materials, 56(20), 3099-3111
Checkerboard:
· Kabir, H., & Aghdam, M. M. (2019). A robust Bézier based solution for nonlinear vibration and post-buckling of random checkerboard graphene nano-platelets reinforced composite beams. Composite Structures, 212, 184-198.
· Kabir, H., & Aghdam, M. M. (2021). A generalized 2D Bézier-based solution for stress analysis of notched epoxy resin plates reinforced with graphene nanoplatelets. Thin-Walled Structures, 169, 108484.
10. Conclusion: Can authors highlight future research directions and recommendations? Also, highlight the assumptions and limitations (e.g. 1-2 shortcoming(s) of the present study). Besides, recheck your manuscript and polish it for grammatical mistakes (you can use “Grammarly” or similar software to quickly edit your document).
Round 2
Reviewer 1 Report
1. Data of graphite must also be plotted in Figure 3 for proper understanding of effectiveness of GO preparation.
2. Correct word is "Hummers' method". kindly recheck the word in your complete file.
3. line 147: what does it mean "32 scanning times". I think it was "scan rate of 32"
4. Abbreviations must be defined in detail at the very first place of their use. e.g. PU has been utilized in line 73 but defined later in line 76.
likewise after defining abbreviation, full form must be avoided, i.e. line 153 "water-based polyurethane" must be "WPU".
Authors must check whole draft for such typo-errors.
Reviewer 2 Report
The authors addressed my comments and the manuscript can be published in the present format
Author Response
Thanks for your suggestion.
Round 3
Reviewer 1 Report
Authors have responded to all comments